# Rapid Self-Assembly of Metal/Polymer Nanocomposite Particles as Nanoreactors and Their Kinetic Characterization

**DOI:** 10.3390/nano9030318

**Published:** 2019-02-28

**Authors:** Andrew Harrison, Tien T. Vuong, Michael P. Zeevi, Benjamin J. Hittel, Sungsool Wi, Christina Tang

**Affiliations:** 1Department of Chemical and Life Sciences Engineering, Virginia Commonwealth University, Richmond, VA 23284-3028, USA; harrisona3@vcu.edu (A.H.); vuongtt@vcu.edu (T.T.V.); zeevimp@vcu.edu (M.P.Z.); hittelbj@vcu.edu (B.J.H.); 2The National High Magnetic Field Laboratory, Florida State University, Tallahassee, FL 32310, USA; sungsool@magnet.fsu.edu

**Keywords:** nanoreactor, catalyst confinement, Flash Nanoprecipitation, diffusion

## Abstract

Self-assembled metal nanoparticle-polymer nanocomposite particles as nanoreactors are a promising approach for performing liquid phase reactions using water as a bulk solvent. In this work, we demonstrate rapid, scalable self-assembly of metal nanoparticle catalyst-polymer nanocomposite particles via Flash NanoPrecipitation. The catalyst loading and size of the nanocomposite particles can be tuned independently. Using nanocomposite particles as nanoreactors and the reduction of 4-nitrophenol as a model reaction, we study the fundamental interplay of reaction and diffusion. The induction time is affected by the sequence of reagent addition, time between additions, and reagent concentration. Combined, our experiments indicate the induction time is most influenced by diffusion of sodium borohydride. Following the induction time, scaling analysis and effective diffusivity measured using NMR indicate that the observed reaction rate are reaction- rather than diffusion-limited. Furthermore, the intrinsic kinetics are comparable to ligand-free gold nanoparticles. This result indicates that the polymer microenvironment does not de-activate or block the catalyst active sites.

## 1. Introduction

Self-assembled amphiphilic molecules, both small molecules and macromolecules, that confine catalysts to micelle, vesicle, and Janus particle nanoreactor systems have proven to offer an efficient approach to perform organic reactions using water as a bulk solvent [1,2,3]. Using Janus particles, catalysts can be incorporated into a portion of the nanocomposite particle and the other portion imparts stability to the system. Asymmetric catalyst loading can facilitate particle motion driven by a chemical reaction [3]. In other nanoreactor systems, surfactant micelles that incorporate organic, metal (homogeneous), and metal nanoparticle catalysts have been used for a wide range of coupling reactions in water [4,5,6]. Confining catalyzed organic reactions to the nanoreactor environment can be leveraged to speed up various chemical reactions [7,8]. Improved yield and selectivity when compared to traditional organic solvents has been reported [6,9].

Core-shell polymer systems have also been considered. Polymeric micelles have been used for several reactions such as asymmetric aldol reactions catalyzed by L-proline [10], acylation [11], hydroaminomethylation of octane catalyzed by Ru-based nanoparticles [12], etc., with extensive reviews available elsewhere [13]. Another approach has been to immobilize metal nanoparticles within polyelectrolyte-brushes synthesized on a polystyrene core [14]. The polymer microenvironment of these systems can lead to increased local concentrations of reactants, which can accelerate reactions, facilitate reactions of otherwise non-reactive species [9,10,15,16,17], confer temperature or pH dependent catalytic activity [18], and/or provide specificity based on hydrophobicity [15].

Generally, these promising approaches have involved design and synthesis of amphiphiles, block copolymers, or polyelectrolytes that contain catalyst or ligand for covalent attachment of the catalyst. Additionally, nanoreactor properties, such as catalyst loading and nanoreactor size, are related to the molecular properties of the synthesized material. Thus, varying the nanoreactor properties would require additional syntheses. Approaches to metal nanoparticle catalyst-polymer nanocomposite particle fabrication that would facilitate (1) modular material (off-the-shelf polymer, catalyst) selection, (2) tunable properties (size and catalyst loading), and (3) rapid, scalable productionwould be beneficial to expanding their potential application.

Flash NanoPrecipitation (FNP) is a rapid, scalable method of polymer self-assembly that may be useful for producing nanoreactors. In Flash NanoPrecipitation, an amphiphilic block copolymer and hydrophobic core material are dissolved in a water miscible organic solvent and rapidly mixed against water using a confined impinging jet mixer. Upon mixing, the rapid decrease in solvent quality causes the hydrophobic core material to precipitate and the block copolymer to micellize directing formation of the overall nanocomposite particle. This particle assembly ends when the hydrophobic block of the block copolymer adsorbs on the precipitating core material preventing further growth, while the hydrophilic block sterically stabilizes the nanoparticle. Given the molecular weight of the block copolymer, dynamic exchange of the block copolymer does not occur [9,19,20], so the resulting structure is kinetically-trapped.

Hydrophobic, inorganic nanoparticles have been incorporated into nanocomposite particles by dispersing the nanoparticles with the dissolved block copolymer and then mixing with confined impinging jets. Upon mixing, colloidal aggregation and block copolymer self-assembly occur due to the decrease in solvent quality. Nanocomposite particle assembly is complete when sufficient hydrophobic blocks of the block copolymer adsorb to the nanoparticle clusters to prevent further colloidal aggregation. For example, Gindy et al. demonstrated fabrication of polymer nanostructures containing colloidal gold using Flash NanoPrecipitation [21]. More recently, Pinkerton et al. encapsulated iron oxide nanoparticles for medical imaging applications [22]. For medical imaging, ~100 nm composite nanostructures with tunable inorganic nanoparticle loading were achieved. These studies suggest that Flash NanoPrecipitation is a suitable method for nanoreactor fabrication. However, the ability to independently tune inorganic nanoparticle loading and nanocomposite particle size has yet to be demonstrated.

Other important considerations when using the nanocomposite particles as nanoreactors are the reaction and diffusion within the system. In small molecule micelle systems that are thermodynamically stable, there is constant molecular exchange between the bulk solvent, and the confined hydrophobic mesophase facilitates reaction [6,9]. In the kinetically-trapped systems produced by Flash Nanoprecipitation, reactants and products reach the catalyst by partitioning from the bulk and diffusing through the nanoreactor structure [17]. The potential mass transfer limitations and the effect of incorporation into the nanocomposite particle on reactivity of the catalyst need to be established.

In this work, we use Flash NanoPrecipitation for rapid and scalable self-assembly of hybrid metal nanoparticle catalyst-polymer nanocomposite nanoreactors. Independently tuning the nanoreactor properties, namely size and gold loading, is investigated. We focus on fundamental understanding of reaction and diffusion using the reduction of 4-nitrophenol as a model reaction. Kinetic and scaling analysis following the induction time are also discussed.

## 2. Materials and Methods

### 2.1. Materials

Citrate stabilized 5 nm gold nanoparticles were purchased from Ted Pella. Polystyrene (PS, M_W_ 800–5000 g/mol) was purchased from Polysciences, Inc. Sodium borohydride and 4-nitrophenol were purchased from Sigma Aldrich (St. Louis, MO, USA). Dodecanethiol (DDT) stabilized 5 nm nanoparticles, tetrahydrofuran (THF), HPLC grade), ethanol (ACS reagent grade), and diethyl ether (ACS reagent grade) were purchased from Fisher Scientific (Fairmont, NJ, USA). Environmental Grade Hydrochloric Acid 30-38% and Environmental Grade Nitric Acid 70% were purchased from GFS Chemicals (Columbus, OH, USA). The ^1^H-NMR solvent D_2_O with 4,4-dimethyl-4-silapentane-1-sulfonic acid DSS as an internal standard was purchased from Cambridge Isotope Lab, Inc (Andover, MA, USA). These chemicals and materials were used as received. Polystyrene-b-polyethylene glycol (PS-b-PEG, PS_m_-b-PEG_n_ where m = 1600 g/mol and n = 5000 g/mol) was obtained from Polymer Source (Product No. P13141-SEO). Prior to use, PS-b-PEG was dissolved in THF (500 mg/mL) and precipitated in ether (~1:20 *v*/*v* THF:ether). The PS-b-PEG was recovered by centrifuging, decanting, and drying under vacuum at room temperature for 2 days.

### 2.2. Nanoreactor Assembly

For self-assembly, the gold nanoparticles need to be dispersed in a water miscible solvent such as THF with molecularly dissolved block copolymer. To disperse the gold nanoparticles in THF, the as-received dodecanethiol stabilized gold nanoparticles in toluene (1 mL) were precipitated into ethanol (45 mL) and filtered using a Buchner funnel. The filtered nanoparticles from the filter cake were resuspended in THF and concentrated via evaporation at room temperature overnight to achieve a nominal concentration of around 20 mg/mL. The final concentration was confirmed by inductively coupled plasma optical emission spectroscopy using an Agilent 5110 (ICP-OES, Santa Clara, CA, USA). UV spectra collected on an Ocean Optics FLAME-S-UV-VIS with a HL-2000-FHSA light source (Largo, FL, USA) were compared before and after the solvent switch to confirm processing did not significantly affect gold nanoparticle size.

For nanoreactor self-assembly, typically, PS-b-PEG (6 mg), dodecanethiol stabilized 5 nm gold nanoparticles (0.5 mg), and PS homopolymer (co-precipitate, 5.5 mg) were added to 0.5 mL of tetrahydrofuran (THF) and sonicated at 55 °C for 30 min. Using a manually operated confined impinging jet mixer with dilution (CIJ-D) [23,24] with achievable Reynolds’ numbers >1300, the resulting THF mixture was rapidly mixed against 0.5 mL of water into a stirring vial of water (4 mL). The resulting dispersion (5 mL total) was stored at room temperature for further characterization and analysis without purification. The nanocomposite particle properties were tuned by adjusting the total solids concentration or the relative amounts of gold nanoparticles and the co-precipitate at a constant total mass or a constant total core volume based on the bulk density of gold and co-precipitate.

### 2.3. Nanoreactor Characterization

Nanoreactor size was measured after mixing using a Malvern Zetasizer Nano ZS (Westborough, MA, USA) with a backscatter detection angle of 173°. Size distributions are reported using the average of four measurements of the intensity weight distributed with normal resolution. The reported size is the peak 1 mean intensity. The polydispersity index (PDI) is defined from the moment of the cumulant fit of the autocorrelation function calculated by the instrument software (appropriate for samples with PDI < 0.3) and is reported as a measure of particle size distribution. UV absorbance spectra (300 to 1200 nm) of the nanoparticle dispersions were measured at room temperature with an Ocean Optics FLAME-S-UV-VIS with a HL-2000-FHSA light source (Largo, FL, USA). For visualization by TEM, samples were submerged in a dilute dispersion of nanoreactors (10-fold dilution with water) for one hour and dried at ambient conditions overnight. Samples were imaged using a Zeiss Libra 120 TEM (Oberkochen, Germany) using an accelerating voltage of 120 kV. To determine the gold nanoparticle concentration, nanoreactor dispersions were dissolved in THF and digested in aqua regia (1:3 nitric acid:hydrochloric acid by volume) and diluted to 5% *v*/*v* aqua regia. Gold concentration of the digested sample was measured using inductively coupled plasma optical emission spectroscopy measurements with an Agilent 5110 (Santa Clara, CA, USA).

### 2.4. Kinetic Analysis

The catalytic performance of the nanoreactors was evaluated using the reduction of 4-nitrophenol with sodium borohydride as a model reaction using well established procedures [25,26]. Briefly, the nanoreactors were diluted with 4-nitrophenol (aq.) and aqueous sodium borohydride (within 5 min of preparation) and the reduction of 4-nitrophenol was monitored using UV spectroscopy (Ocean Optics FLAME-S-VIS-NIR-ES, Largo, FL, USA, with a HL-2000-FHSA light source (300–1200 nm) with a CUV-UV cuvette holder placed on a stir plate). The final reaction mixture contained less than 0.01 vol% THF. The induction time and apparent reaction rate (*k_app_*) were determined from tracking the absorbance at 425 nm as a function of time. The values of *k_app_* and induction time are the averages (± standard deviations) of at least 3 trials of each experiment. Detailed procedures are provided in the Appendix A.

### 2.5. Langmuir-Hinshelwood Kinetics

For more detailed kinetic analysis, we performed full kinetic analysis considering the two-step reaction mechanism previously established [26]. Full kinetic analysis is described by the reaction rate of each step and the Langmuir adsorption constants of 4-nitrophenol, borohydride, and the stable intermediate. We determined the rate constants for both steps by solving the coupled rate equations using the numerical method previously described and fitting the experimental data (average of three experimental trials) [26].

### 2.6. NMR Measurements

To evaluate effective transport of the 4-nitrophenol, ^1^H-NMR spectroscopy and pulsed field gradient (PFG) NMR, combined with saturated transfer difference (STD) spectroscopy, using a Bruker 800 MHz cryo-probe (Billerica, MA, USA) was performed in accordance with the methods described in the Appendix A. Briefly, 4-nitrophenol molecules in close proximity to the nanoreactor core were analyzed based on spin diffusion of selectively saturated polystyrene, in conjunction with an applied magnetic field gradient. Relevant intensities were analyzed as a function of gradient strength to determine the diffusion coefficient of the molecules. Since nanoreactors diffuse in free solution at least 3 orders of magnitude slower than molecules, the measured diffusion coefficient was considered the effective diffusion coefficient of the solute within the nanoreactor [27,28,29].

## 3. Results and Discussion

### 3.1. Nanoreactor Self-Assembly

To perform Flash NanoPrecipitation, dodecanethiol stabilized 5 nm gold nanoparticles were dispersed in THF with the molecular dissolved, PS and PS-b-PEG, and rapidly mixed with water using a hand-operated confined impinging jet mixer. The entire formation process was accomplished in less than a second; further, the process can be performed continuously at large scales [24,30,31]. Due to their hydrophobic nature and particle aggregation during assembly, the gold nanoparticles are expected to be in the hydrophobic core of the nanoreactor [3,21], forming a nanoparticle-macromolecular system [32]. Due to the high molecular weight of the polystyrene block, no dynamic exchange of the block copolymer is expected [20]. The resulting nanoreactors were ~130 nm indicated by a single Gaussian peak with PDI <0.2 measured by dynamic light scattering (DLS). The dispersions were stable when stored at room temperature for at least 2 months as there was no significant change in size or size distribution by DLS (Appendix A), and no macroscopic precipitation of unencapsulated gold was observed.

We further characterized the nanoreactors using UV-Vis spectroscopy. Prior to Flash NanoPrecipitation, the dodecanethiol-stabilized nanoparticles dispersed in toluene showed a peak absorbance at 495 nm (as received and after switching solvents). The nanoreactors showed a peak absorbance of 520 nm (Figure 1b). The peak shift could occur due to differences in hydrophobicity of the surrounding environment [33]. Since the polystyrene microenvironment should have similar hydrophobicity as toluene, we attribute the red-shift to plasmonic coupling due to close proximity of the encapsulated gold nanoparticles, which has been previously observed with polymer-gold nanocomposite particles [34].

The structure of the nanocomposite particles was visualized using TEM. Based on TEM imaging, clustering of the gold nanoparticles during assembly resulted in multiple catalytic gold nanoparticles per nanoreactor. The majority of the gold nanoparticles appear to be in the nanoreactor core, although multiple polymer layers are not visible on TEM due to low electron density. This result is consistent with previous reports of encapsulated gold nanoparticles via Flash NanoPrecipitation (FNP) [21,35]. Based on TEM, some of the gold may also be associated with the PEG-layer of the nanoreactors whereas unassociated gold would be expected to precipitate out of the dispersion as well as affect the size distribution measured by DLS. Since we do not observe gold precipitate from the dispersion, and the size of the TEM size is consistent with DLS with PDI <0.2, we assume all the gold in the dispersion is associated with the nanoreactors. Finally, we confirmed the amount of gold by ICP-OES. We found the polystyrene nanoreactors retained 74% of the gold from the THF-gold nanoparticle solution and the loss can be attributed to the hold-up volume during mixing.

Next, we aimed to independently tune the nanoreactor properties, size and gold loading, using formulation parameters. Nanoreactor assembly depends on the relative time scales of block copolymer micellization, gold nanoparticle clustering, and co-precipitate nucleation and growth. Therefore, the overall nanoreactor size can be affected by the ratio of core material to block copolymer, as well as the total concentration of components in the organic stream [22].

Varying the ratio of block copolymer to core materials has been an effective method for tuning nanostructure size via Flash NanoPrecipitation [23,36]. To vary nanoreactor size, the amount of block copolymer concentration can be increased (Appendix A), but the gold loading is also affected. In order to vary the nanoreactor size while holding the gold loading constant, we varied the total solids concentration holding the mass ratio of gold to polystyrene co-precipitate constant. As expected, the nanoreactor size increased with increasing total solids concentration. This effect has been attributed to an increase in the rate of particle core relative to nucleation [36,37]. Using this approach, the nanoreactor size could be tuned between 100 and 200 nm with nominal gold loading of 4 wt % (Figure 2a). This level of gold loading is comparable other polymer nanocomposite systems with low volume additions of inorganic nanoparticles that demonstrate enhanced functional performance [32].

Next, we aimed to vary the gold loading independently of nanoparticle size. Holding the total core material mass constant and varying the ratio of gold to polymer resulted in a decrease in nanoparticle size with increasing gold concentration. In contrast, with gold nanoparticles and block copolymer without a co-precipitate, Gindy et al. observed that increasing the gold loading results in an increase in nanocomposite particle size that is attributed to the increase in the amount of gold core relative to the block copolymer [21]. The difference is our use of a co-precipitate. We attribute the trend observed in this case to the increase in the number density of gold nanoparticles that act as nucleating agents that seed particle growth via heterogeneous nucleation [37,38].

To guide nanoreactor formulation, the Smoluchowski diffusion limited aggregation model has previously been used to formulate inorganic nanoparticle-polymer nanocomposite particles via Flash NanoPrecipitation [36]. Based on the model, nanoreactor size can be predicted using:(1)$$R={\left(K\frac{k_{B}Tc_{core}^{5/3}}{\pi \mu \rho c_{BCP}}\right)}^{1/3}$$
where *R* is the aggregate radius, *K* is a constant of proportionality for formation time, *k_B_* is Boltzmann’s constant, *T* is the absolute temperature, *c_core_* is the concentration of core material, *c_BCP_* is the concentration of block copolymer, *µ* is the solvent viscosity, and *ρ* is the core material density. This model suggests that the nanoreactor size is affected by the volume more than the mass of the core. Thus, as an alternative to holding the mass of the core constant, we held the volume of the core constant, according to:(2)$$V_{cm}=\frac{m_{AuNP}}{\rho_{AuNP}}+\frac{m_{PS}}{\rho_{PS}}$$
where $V_{cm}$ is the total volume of the core materials, $m_{AuNP}$ and $m_{PS}$ are the masses of the gold nanoparticles and polystyrene core materials, respectively, finally $\rho_{AuNP}$ and $\rho_{PS}$ are the densities of the gold nanoparticles and polystyrene core materials, respectively. The core volume was selected from the standard formulation, a nominal gold loading of 4% and nanoreactor concentration of 2.4 mg/mL. Using the density of bulk gold and polystyrene, which are 19.32 g/mL and 1.04 g/mL, respectively, the core material volume was found to be 5.33 µL. Using the approach of constant volume, the gold loading was tuned between 4 and 50 nominal wt % at a nanoreactor size of ~130 nm (Figure 2b).

Overall, nanoreactors were assembled in a rapid, scalable, single-step method using Flash NanoPrecipitation. Nanoreactor size could be tuned independently of gold nanoparticle loading by varying the total solids concentration at a constant ratio of gold to polystyrene. Interestingly, the gold nanoparticle loading was tuned independently of nanoreactor size by varying the ratio of gold to polystyrene at constant total core volume. The constant core volume approach may be useful for formulations of multiple components with disparate densities e.g., inorganic particle-polymer nanocomposite particles.

### 3.2. Initial Characterization of Nanoreactor Performance

To evaluate the catalytic performance of the nanoreactor, the reduction of 4-nitrophenol by sodium borohydride was used as a model reaction [39]. First, we confirmed the nanoreactors remained intact following the reaction; no significant change in size or polydispersity was observed by DLS (Figure 1a). Further, no macroscopic precipitation of gold nanoparticles was observed following the reaction.

In these initial studies, we assume all of the gold nanoparticles included in the formulation are associated with the nanoreactor and contribute to the observed catalytic activity. From TEM (Figure 1c), the gold nanoparticles may be associated with the hydrophobic core or hydrophilic shell or may be unencapsulated. Unencapsulated gold was not observed precipitating from the nanoreactors and would not contribute to the observed activity (Table 1, dodecanethiol-stabilized gold nanoparticles (DDT)). This is likely due to the lack of solubility as other hydrophobic inorganic nanoparticles have shown activity in water:solvent reaction mixtures [40]. If the dispersions contained trace amounts of unencapsulated gold, the reported values for k_1_ would be slightly underestimated. The conversion of 4-nitrophenol confirmed the gold nanoparticles associated with the nanoreactors were catalytically active (Appendix A). The apparent reaction rate constant per surface area of gold, k_1,_ for the nanoreactors was 0.414 ± 0.095 L m^−2^s^−1^, which is comparable to the citrate-stabilized, 5 nm gold particles.

Comparing the performance of the nanoreactors with other metal nanoparticle-polymer systems using the reaction rate considering the amount of gold catalyst (e.g., k_1_ in Table 2), the nanoreactors demonstrate over 110-fold better catalytic activity than gold within polymer (PNIPAM-b-P4VP) micelles, despite a larger overall nanoreactor size. This difference may be attributed to P4VP-gold interactions that affect availability of active sites. Thus, the use of non-interacting co-precipitates and Flash NanoPrecipitation may provide an advantage to other polymer micelle systems that rely on gold-polymer interactions for self-assembly.

Further, the induction time and kinetics are similar to immobilized gold nanoparticles within polyelectrolyte brush shell on polystyrene core particle systems [26]. Specifically, the kinetics of the nanoreactors we report with 5 nm gold are comparable to polyelectrolyte brushes with 2.2 nm gold nanoparticles at the surface of the core-shell nanostructures, which are expected to have similar activities [42]. This result suggests that association of the catalyst with the nanoreactor does not sacrifice reactivity.

### 3.3. Probing Potential Mass Transfer Limitations

#### 3.3.1. Induction Time

Notably, the induction time of the encapsulated gold nanoparticles is ~50-fold longer than citrate-stabilized nanoparticles (Table 1). This relatively long induction time has been previously observed with gold-nanoparticle-polymer nanoreactor systems. It may be attributed, in part, to slow surface restructuring upon encapsulation within the hydrophobic polystyrene microenvironment [14]. Additional factors that may increase induction time include: poisoning of the active sites when encapsulated within the nanoreactor core, reduction of the dissolved oxygen present in the reaction dispersion, and/or diffusion limitations [44,45].

To further understand the nature of the induction time in the nanoreactor system, we investigated both the sequence of addition and the time between adding the reactants (Figure 3). Under standard model reaction conditions, 4-nitrophenol was added first and allowed to equilibrate for 1 min, followed by the addition of the sodium borohydride. To probe potential diffusion limitations, we increased the time between adding the 4-nitrophenol and sodium borohydride 10-fold, and no significant change in induction time was observed. This result suggests that the induction time is not related to diffusion of 4-nitrophenol.

Moreover, switching the sequence to adding sodium borohydride first, followed by 4-nitrophenol after 1 min of equilibration did not significantly affect the induction time. Interestingly, when the equilibration time was increased in this case, the induction time was reduced by two orders of magnitude. This ~5 s induction time is comparable to the value measured for citrate-capped gold nanoparticles. This result indicates the long induction times relative to citrate stabilized gold nanoparticles may be attributed to diffusion of sodium borohydride. Further examining the effect of equilibration time, the induction time decreased from ~100 to 5 s when increasing the equilibration time from 1 to 3 min (Figure 4). Further increasing the equilibration time beyond 3 min did not significantly impact the induction time. Thus, it appears that it takes ~3 min for sufficient sodium borohydride to partition into the nanoreactor for the reaction to progress. This required equilibration time can be reduced by increasing the concentration of the borohydride (constant ratio of borohydride to 4-nitrophenol) (Appendix A) which further indicates the relatively long induction time of the nanoreactors relative to citrate stabilized gold nanoparticles can be attributed to diffusion of the borohydride.

#### 3.3.2. Reaction Rate

Next, we further investigated potential mass transfer limitations on the observed reaction rate following the induction time. A useful tool for determination of diffusion limitations is the second Damköhler number (DaII), which is a ratio of the reaction rate to the diffusion rate given by:(3)$$DaII=\;\frac{k_{app}C^{n-1}}{\beta a}$$
where *n* is the reaction order, β is the mass transport coefficient (which is a quotient of the diffusion coefficient and the characteristic length of the system), and a is the interfacial area. To calculate DaII for a 130 nm diameter particle, the interfacial area (nanoreactor area per unit volume of nanoreactor dispersion) was estimated to be 2 × 10^4^ m^−1^ based on the number of nanoreactors estimated using the aggregation number of the block copolymer previously reported [46,47]. The diffusion coefficient for 4-nitrophenol in the nanoreactor system was experimentally determined by NMR. Using PFG-NMR in conjunction with the STD spectroscopy, the effective diffusion coefficient of the 4-nitrophenol within the nanoreactors was determined to be 1.91 ± 0.01 × 10^−8^ m^2^/s (Appendix A). Using this experimentally determined effective diffusion coefficient, the DaII is on the order of 10^−6^ indicating the reaction is significantly slower than diffusion; therefore, the apparent kinetics are reaction-limited.

A complementary approach was to consider the bimolecular reaction between 4-nitrophenol and nanoparticle catalyst using the Smoluchowski diffusion limited reaction model [48,49]. We varied the gold concentration by (1) varying the nanoreactor concentration to probe potential external diffusion limitations, and (2) varying the gold loading at constant nanoreactor concentration to examine potential internal diffusion limitations (Appendix A). When the nanoreactor concentration or the gold loading was increased, *k_app_* increased; the 2nd order rate constant was on the order of 10^6^ M^−1^s^−1^. These values are much lower than the k_bm_ ~ 10^8^ M^−1^s^−1^, indicating that neither internal nor external diffusion from the bulk solution to the nanoreactor limited the apparent reaction kinetics.

Since there were no indications of diffusion limitations associated with the reaction following the induction time, we further characterized the reaction kinetics using Langmuir-Hinshelwood kinetics. Based on the previously established two-step reaction model [26], and fitting the measured concentration of 4-nitrophenol as a function of time (normalized after the induction time for conversions up to 30%) [26], the kinetics were comparable to other gold nanoparticle-polymer nanoreactor systems (Table 3 with plot, Appendix A, and full fit parameters, Appendix A). Interestingly, k_a_ and k_b_ observed for the gold encapsulated within the nanoreactors are comparable to ligand-free gold nanoparticles. This result suggests that the reactivity of the gold nanoparticle surface is not significantly affected by self-assembly and their incorporation into the nanoreactors.

Overall, diffusion and partitioning of sodium borohydride into the polymer nanoreactor affect the induction time for the reaction. Sufficient equilibration time between adding the sodium borohydride and the 4-nitrophenol (~3 min) for the borohydride to partition and diffuse minimizes induction time. Notably, mass transfer effects are not observed after the induction time and the intrinsic kinetics are comparable to ligand-free gold nanoparticles.

## 4. Conclusions

Overall, we have presented rapid, scalable self-assembly of metal nanoparticle catalyst-polymer nanocomposite particles as nanoreactors. The size and gold loading of the nanoreactors can be tuned independently, with sizes and nominal loadings ranging from 100–200 nm and 4–50 wt% respectively. Using the 4-nitrophenol reduction as a model reaction, the induction time is affected by sequence or reagent addition, time between addition, and reagent concentration. Combined, our experiments indicate that the induction time is most influenced by diffusion of sodium borohydride. Scaling analysis and effective diffusivity measured using NMR, the observed reaction rate after the induction time are reaction- rather than diffusion-limited. Finally, the intrinsic reaction kinetics of gold associated with the polymer were comparable to ligand-free particles indicating the self-assembly process and resulting polymer microenvironment did not de-activate or block the catalyst active sites. Building on this foundational study, practical considerations such as nanoreactor recycling will be considered in future work.

## Figures and Tables

**Figure 1 nanomaterials-09-00318-f001:**
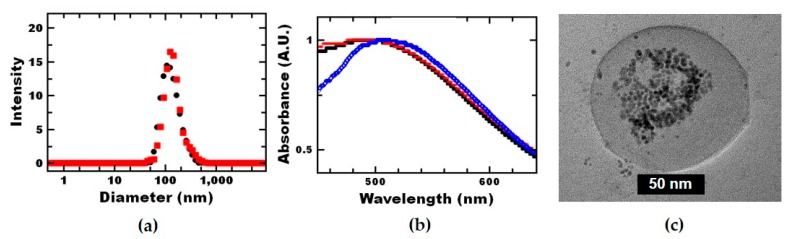
Polymer nanoreactors were fabricated via self-directed assembly. (**a**) DLS confirms the uniform size distribution of the ~130 nm self-assembled polymer nanoreactors (black circles) and confirms that the size is the same after the reduction of 4-nitrophenol (red squares). (**b**) UV-vis analysis shows that the absorbance of the gold nanoparticle remains unchanged through the solvent switch from toluene (black filled circles) to tetrahydrofuran (THF) (red open circles). A red-shift is seen upon encapsulation within polymer nanoreactors (blue open diamonds) due to close proximity of the encapsulated gold nanoparticles. (**c**) TEM imaging demonstrates that multiple gold nanoparticles were encapsulated within the core of the nanoreactors.

**Figure 2 nanomaterials-09-00318-f002:**
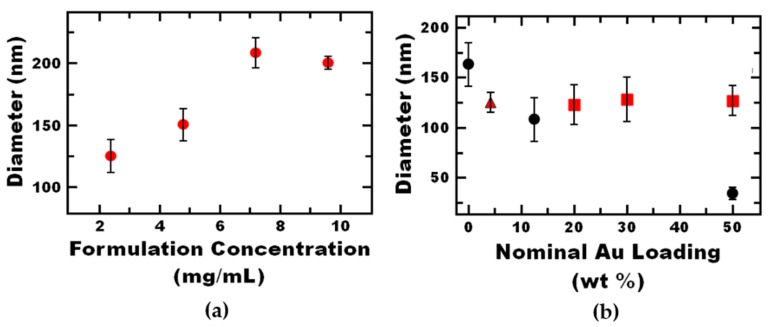
Hydrodynamic diameter of polystyrene nanoreactors measured by DLS with varying total nanoreactor material concentration in the formulation. (**a**) By varying the total material concentration with constant ratio of components tunable nanoreactor size between 100–200 nm. (**b**) By varying the gold to polystyrene co-precipitate ratio at a constant nanoreactor core volume (red squares), as opposed to constant mass ratio (black circles), the nominal gold loading of polystyrene nanoreactors can be tuned at constant nanoreactor size (~130 nm). The standard formulation (4 wt % nominal gold loading, 2.4 mg/mL) is shown by the red triangle.

**Figure 3 nanomaterials-09-00318-f003:**
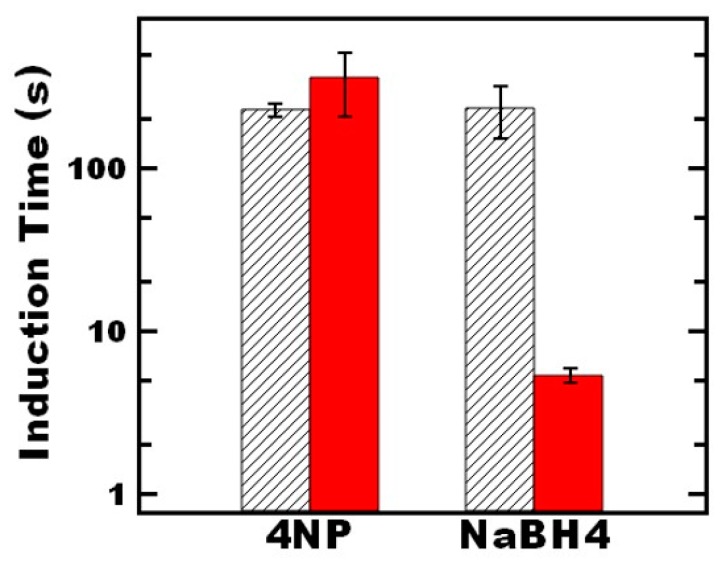
The effect of the sequence of reagent addition on the induction time of the 4-nitrophenol reaction. In all experiments, the 4-nitrophenol and sodium borohydride concentration followed standard conditions of 0.01 mM and 0.01 M, respectively. The indicated reagent was the first to be added, after which the reagent was allowed to equilibrate in the solution for either 1 min (black striped bars) or 10 min (red solid bars). The end of the equilibration period was the addition of the second reagent, at which point the reaction could progress.

**Figure 4 nanomaterials-09-00318-f004:**
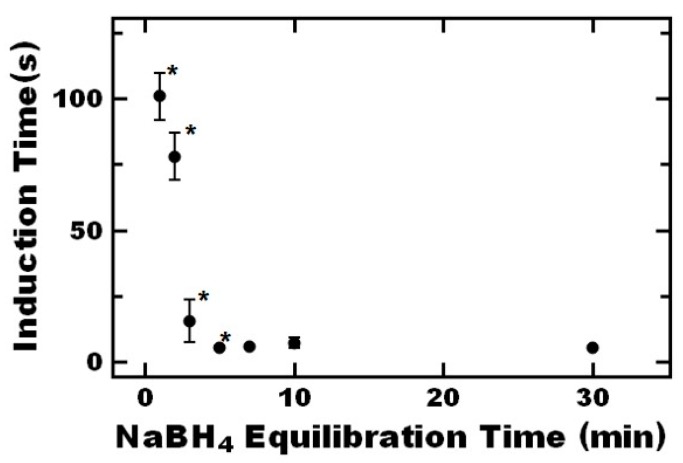
The effect of sodium borohydride equilibration on the induction time of the 4-nitrophenol reaction. Standard reagent concentrations of 0.01 mM and 0.01 M were used for 4-nitrophenol and sodium borohydride, respectively. Data points marked with an asterisk (*) are significantly different than each other (*p* < 0.1).

**Table 1 nanomaterials-09-00318-t001:** Rate constants and induction times for various gold nanoparticles. PS, polystyrene; DDT dodecanethiol-stabilized gold nanoparticles.

Support	Diameter (nm)	k_1_ (L m^−2^ s^−1^)	Induction Time (s)	Reference
PS	5	0.414 ± 0.095	229 ± 21	This Paper
DDT	5	Undetected	N/A	This Paper
Citrate	5	0.173 ± 0.026	5 ± 1	This Paper
Ligand-Free	7	0.17	N/A	[41]

**Table 2 nanomaterials-09-00318-t002:** Rate constants for various metal/polymer nanocomposite nanoreactors.

Support	AuNP Diameter (nm)	k_1_ (L m^−2^ s^−1^)	Reference
Polystyrene nanoreactors	5	(4.14 ± 0.95) × 10^−1^	This Paper
PNIPAM-b-P4VP Micelles	3.3	3.70 × 10^−3^	[43]
Polyelectrolyte brush	2.2	2.70 × 10^−1^	[14]

**Table 3 nanomaterials-09-00318-t003:** Langmuir-Hinshelwood rate constants obtained from fits to experimental data.

Reactor	k_a_ (10^4^ mol/m^2^ s)	k_b_ (10^5^ mol/m^2^ s)	Reference
Polystyrene Nanoreactors	4.32 ± 0.14	4.3 ± 0.5	This Study
Ligand-Free	5.8 ± 3.1	5.4 ± 2.0	[41]
Brush Shell	9.7 ± 2.9	7.8 ± 1.7	[26]

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
