# Peer review of "Rapid Self-Assembly of Metal/Polymer Nanocomposite Particles as Nanoreactors and Their Kinetic Characterization"

_nanomaterials, 2019, doi:10.3390/nano9030318_

Reviewer 1 Report

The manuscript entitled “Rapid Self-Assembly of Metal/Polymer Nanocomposite Particles as Nanoreactors and their Kinetic Characterization” has demonstrated rapid, scalable self-assembly of metal nanoparticle catalyst polymer nanocomposite particles via Flash NanoPrecipitation. Furthermore, nanocomposite particles have been utilized as nanoreactors and demonstrated for the reduction of 4-nitrophenol as a model reaction. According to this reviewer, the present work is good and could be beneficial for the researcher working in this field.

The following comments/suggestions should be incorporated before acceptance in its current format.

They are:

Please check the manuscript for typographical errors.

Some relevant references are missing on the synthesis of nanocomposite. For better understanding of the review the following references should be cited and discussed in the main body of the manuscript (Scientific reports 2017, 7, 16311; Polymers 2016, 8, 105; Chem. Mater., 2007, 19, 2736–2751; Chem. Rev., 2013, 113, 5194–5261).

Reviewer 2 Report

The systems studied in the paper are interesting and the approaches used by the authors are justified, while the results are confirmed by substantial experimental data. The following comments should be taken into account in the revision of the paper:

There is a duplication of the rate constants or induction periods in Tables 1,2 and figures 2,3.

It is preferable to present the activity as rate constants at a low conversion, as the authors did (at 15% conversion), however, the conversion vs. time dependence presentation in SI would give a general estimate of the reaction performance.

Line 188, "absorbance of 520 nm (Figure 1(a))" is not correct. It should be: "absorbance of 520 nm (Figure 1(b))."

Figure 1a is exactly the same as Fig. S5. One of them can be removed.

Reviewer 3 Report

The manuscript by Tang and collaborators describe the preparation of self-assembled metal polymer nanocomposite nanoparticles as nanoreactors using a promising approach called flash nano-precipitation for reactions in water like p-nitrophenol reduction with sodium borohydride.

The work has been carefully carried out, the paper is also well written describing the objectives of the research.

Overall the manuscript presents sufficient degree of novelty for publication in Nanomaterials.

I consider the paper worth of publication, basically in its present form.

Just a very minor issue:

-table 3 please correct the value 4.31 ± 0.5, as 4.3 ± 0.5
